# Mucins 3A and 3B Are Expressed in the Epithelium of Human Large Airway

**DOI:** 10.3390/ijms241713546

**Published:** 2023-08-31

**Authors:** Heta Merikallio, Terezia Pincikova, Ioanna Kotortsi, Reza Karimi, Chuan-Xing Li, Helena Forsslund, Mikael Mikko, Sven Nyrén, Elisa Lappi-Blanco, Åsa M. Wheelock, Riitta Kaarteenaho, Magnus C. Sköld

**Affiliations:** 1Research Unit of Biomedicine and Internal Medicine, University of Oulu, 90570 Oulu, Finland; heta.merikallio@oulu.fi (H.M.);; 2Center of Internal Medicine and Respiratory Medicine, Medical Research Center Oulu, University Hospital of Oulu, 90220 Oulu, Finland; 3Respiratory Medicine Unit, Department of Medicine Solna, Center for Molecular Medicine, Karolinska Institutet, 171 77 Stockholm, Sweden; 4Stockholm CF-Center, Albatross, K56, Karolinska University Hospital Huddinge, 141 86 Stockholm, Sweden; 5Department of Respiratory Medicine and Allergy, Karolinska University Hospital, 171 77 Stockholm, Sweden; 6Department of Molecular Medicine and Surgery, Division of Radiology, Karolinska Institutet, Karolinska University Hospital Solna, 171 76 Stockholm, Sweden; 7Cancer and Translational Medicine Research Unit, Department of Pathology, University Hospital of Oulu, Oulu University, 90220 Oulu, Finland

**Keywords:** bronchoalveolar lavage, COPD, epithelium, immunohistochemistry, large airways, lung function, microarray, mRNA, mucin, smoking

## Abstract

Aberrant mucus secretion is a hallmark of chronic obstructive pulmonary disease (COPD). Expression of the membrane-tethered mucins 3A and 3B (MUC3A, MUC3B) in human lung is largely unknown. In this observational cross-sectional study, we recruited subjects 45–65 years old from the general population of Stockholm, Sweden, during the years 2007–2011. Bronchial mucosal biopsies, bronchial brushings, and bronchoalveolar lavage fluid (BALF) were retrieved from COPD patients (n = 38), healthy never-smokers (n = 40), and smokers with normal lung function (n = 40). Protein expression of MUC3A and MUC3B in bronchial mucosal biopsies was assessed by immunohistochemical staining. In a subgroup of subjects (n = 28), MUC3A and MUC3B mRNAs were quantified in bronchial brushings using microarray. Non-parametric tests were used to perform correlation and group comparison analyses. A value of *p* < 0.05 was considered statistically significant. MUC3A and MUC3B immunohistochemical expression was localized to ciliated cells. MUC3B was also expressed in basal cells. MUC3A and MUC3B immunohistochemical expression was equal in all study groups but subjects with emphysema had higher MUC3A expression, compared to those without emphysema. Smokers had higher mRNA levels of MUC3A and MUC3B than non-smokers. MUC3A and MUC3B mRNA were higher in male subjects and correlated negatively with expiratory air flows. MUC3B mRNA correlated positively with total cell concentration and macrophage percentage, and negatively with CD4/CD8 T cell ratio in BALF. We concluded that MUC3A and MUC3B in large airways may be a marker of disease or may play a role in the pathophysiology of airway obstruction.

## 1. Introduction

Increased mucus secretion and accumulation in the airway lumen are clinical hallmarks of various lung diseases such as asthma, cystic fibrosis, and chronic obstructive pulmonary disease (COPD). Many patients with these diseases exhibit increased mucin secretion and mucus hyper-concentration, which leads subsequently to an impaired mucus clearance, persistent neutrophil airway inflammation and bacterial colonization [1], and symptoms of chronic bronchitis. These subjects experience also more exacerbations [2], accelerated lung function decline, and impaired survival [3]. 

Chronic obstructive pulmonary disease (COPD) remains a significant burden and cost to health-care systems, as well as a substantial source of amenable mortality [4], as it is the third leading cause of death worldwide [5]. It is a heterogeneous lung condition characterized by chronic respiratory symptoms due to abnormalities of the airways (bronchitis, bronchiolitis) and/or alveoli (emphysema) that cause persistent, often progressive, airflow obstruction [6]. Increasing importance is placed on attempts to slow down COPD lung disease progression by improving risk factors. Globally, male sex, smoking, age, body-mass index of less than 18.5 kg/m^2^, biomass exposure, and occupational exposure to dust or smoke are all substantial risk factors for COPD [7]. 

The leading etiological factor causing COPD is cigarette smoking [8]. It is estimated that the global prevalence of COPD is 10.3%, and it is appreciably higher in smokers and ex-smokers compared to non-smokers [6]. Smoking alters bronchial-mucus-barrier cell composition and transcriptome and increases mucus production [9]. Thus, in smokers, goblet cell hyperplasia and hypertrophy as well as abnormal patterns of mucin gene expression have been described [10]. In asthmatic patients, exposure to tobacco smoke results in oxidant/antioxidant imbalance, which leads to increases in pro-inflammatory cytokines such as TNFα, IL-6, and IL-8 [11]. Active and second-hand smoke exposure can be quantified by measuring serum and urinary cotinine and exhaled carbon monoxide [12].

Despite progress in the treatment of symptoms and the prevention of acute exacerbations, few advances have been made to ameliorate lung disease progression in COPD. To be able to achieve that, a better understanding of the complex disease mechanisms resulting in COPD, as well as a search for biomarkers, are needed [13]. The 6 min walk distance, heart rate, C-reactive protein, fibrinogen, and white cell count were found to be associated with clinical outcomes in patients with stable COPD [14], with a spillover effect of inflammatory response proposed as the underlying mechanism [15]. COPD patients with emphysema presented increased systemic oxidative stress markers and fibrinogen, compared to COPD patients without emphysema [16]. Moreover, elevated plasma level of Pentraxin 3 was associated with emphysema and mortality in smokers [17]. However, data on airway-derived disease biomarkers in COPD are largely lacking.

The major macromolecular components of mucus are mucin glycoproteins (MUCs). They play important roles in the protection of epithelium, signal transduction, and modulation of the immune system. Airway protein expression is predominated by four membrane-associated mucins (MUC1, MUC4, MUC16, MUC20), one secreted non-gel-forming mucin (MUC7), and two gel-forming mucins (MUC5AC and MUC5B) [18]. Long-term smoking induces enhanced MUC5AC expression [19]. Smoking history and the presence of chronic bronchitis, regardless of airway obstruction, affect both cellular and soluble MUC1 in human airways [20]. In line with that, Kato et al. showed that MUC1 contributed to goblet cell metaplasia and enhanced MUC5AC expression in response to cigarette smoke in vivo [21]. In asthma, the expression of MUC5AC increased remarkably by asthma severity; also, it was associated with airway wall thickness [22]. In ovalbumin-induced asthmatic mice, the key circadian rhythm gene, BMAL1, caused periodic changes in airway MUC1 expression [23]. In addition, protein levels of MUC5AC and MUC8 were significantly elevated by lipopolysaccharide or IL-17A stimulation in human middle ear epithelial cells [24]. COPD exacerbations are often associated with increased sputum production and infection with *Haemophilus influenzae*. Interestingly, *Haemophilus influenzae* has been found to upregulate the MUC5AC gene via TLR2-MyD88-dependent p38 mitogen-activated protein kinase pathway [25]. Moreover, neutrophil elastase induces MUC5AC gene expression by an oxidant-dependent mechanism [26], and cytokines, oxidative stress, and endoplasmic reticulum stress stimulate mucus production [27,28,29]. A recent study indicated that the glycosylated extracellular domains of different transmembrane mucins might have protective functions in respiratory cells by restricting SARS-CoV-2 binding and entry [30], and MUC4 was found to be a cellular biomarker of necrotizing bronchiolitis in influenza A virus infection [31]. Taken together, these studies indicate that mucins could be potential biomarkers of airway diseases.

MUC3A and MUC3B are membrane-tethered mucins that were originally described as being expressed in the intestinal epithelium [32]. Non-synonymous single-nucleotide polymorphisms of MUC3A, involving a tyrosine residue with a proposed role in cell signaling, were reported to confer genetic predisposition to Crohn’s disease [33]. Dohrman et al. studied specifically MUC3 mRNA distribution in human bronchus, using an mRNA in situ hybridization technique, and found that it occurred in a diffuse pattern throughout the bronchial epithelium. However, they could not determine whether the labelling was associated with goblet, ciliated, or basal cells [34]. To our knowledge, MUC3A and MUC3B immunohistochemical expression in human airway epithelium in states of health and disease is largely unknown.

Therefore, the aim of this study was to determine the immunohistochemical MUC3A and MUC3B expression and mRNA levels by investigating bronchial mucosal biopsies and bronchial brushing samples of large airway epithelium in never-smokers, current smokers with normal lung function, and in COPD patients. Moreover, we aimed to assess the biomarker potential of MUC3A and MUC3B expression and of their mRNA levels in large airway epithelium.

## 2. Results

### 2.1. Demographics of the Study Groups

The four study groups were matched in terms of age, sex, smoking history (>10 pack-years), and current smoking habits (smoking > 10 cigarettes/day past 6 months) (Table 1). 

### 2.2. Immunohistochemical MUC3A and MUC3B Expression in Airway Epithelium

Immunohistochemical MUC3A expression was localized to ciliated cells in 86 out of 96 biopsy samples (90%) and it was found in basal cells in four out of 96 biopsy samples (4%). The intensity of MUC3A expression in ciliated cells varied from weak (100) to strong (300). MUC3B expression was localized to both basal cells and ciliated cells in the majority of the biopsy samples (99% and 99% of biopsies, respectively), and its intensity varied from weak (100) to very strong (400) (Figure 1). 

Never-smokers, current smokers with normal lung function, currently smoking COPD patients, and COPD patients who were ex-smokers did not differ in MUC3A expression in ciliated cells. Neither were there any differences in MUC3B expression in basal cells or in ciliated cells (Figure 2A–C). Patients with emphysema had higher MUC3A expression in ciliated cells (Figure 2D), compared to patients without emphysema, but the two groups did not differ in MUC3B expression. COPD diagnosis, smoking history, sex, and chronic bronchitis were neither associated with MUC3A nor with MUC3B expression. However, the four biopsy samples where MUC3A expression was detected in basal cells were all retrieved from female subjects.

MUC3A expression in ciliated cells was negatively correlated with macrophage percentage in BALF (Figure 2E). MUC3B immunohistochemical expression in basal cells was positively correlated with epidermal growth factor receptor (EGFR) expression in basal cells (Figure 2F). MUC3B expression in ciliated cells was strongly intercorrelated with MUC3B expression in basal cells (Appendix A). In ciliated cells, MUC3A and MUC3B expression were not correlated (Appendix A). MUC3A and MUC3B immunohistochemical expression did not correlate with expiratory flows. 

### 2.3. MUC3A and MUC3B mRNA Levels across the Study Groups

Current smokers with normal lung function had higher MUC3A mRNA level in bronchial brushings than never-smokers (Figure 3A). In addition, current smokers with normal lung function and current smoking COPD patients had higher MUC3B mRNA levels than never-smokers (Figure 3B). Male subjects had higher MUC3A and MUC3B mRNA concentration in bronchial brush samples, as compared with female subjects (Figure 3C,D). Importantly, this difference could have been due to a difference in smoking habits, as 73% of the male subjects who provided mRNA were current smokers, while only 54% of the female subjects who provided mRNA were current smokers (Table 2). However, even when including only current smokers in the analysis, the observed gender difference persisted. MUC3A mRNA level was positively correlated with MUC3B mRNA level (Figure 3E). 

In all subjects, MUC3A and MUC3B mRNA levels correlated positively with exhaled carbon monoxide measured before bronchoscopy (Figure 3F,G). However, in the subgroup of current smokers only, MUC3A and MUC3B mRNA did not correlate with exhaled carbon monoxide measured, and COPD diagnosis was associated neither with MUC3A mRNA level nor with MUC3B mRNA level. 

### 2.4. Correlations of MUC3A and MUC3B mRNA Levels with Clinical Parameters and BALF

MUC3B mRNA correlated positively with total cell concentration and macrophage percentage in BALF (Figure 4A,B), and negatively with lymphocyte and neutrophil percentage in BALF, as well as with CD4/CD8 T cell ratio in BALF (Figure 4C–E). In the subgroup of current smokers only, MUC3B mRNA correlated negatively with neutrophil percentage in BALF.

MUC3A mRNA levels in samples acquired by bronchial brushing were negatively correlated with postbronchodilator forced vital capacity (FVC) (Figure 5A). In addition, MUC3B mRNA correlated negatively with postbronchodilator forced expiratory volume in one second (FEV_1_) and positively with residual volume (RV) (Figure 5B,C). Subjects with emphysema or chronic bronchitis did not differ in MUC3A or MUC3B mRNA levels from subjects without emphysema or chronic bronchitis, respectively.

## 3. Discussion

In the present study, we investigated the immunohistochemical expression and mRNA levels of MUC3A and MUC3B in human large airway epithelium. Here, we showed for the first time that the immunohistochemical expression of both MUC3A and MUC3B in the large airway epithelium was specifically localized to ciliated cells, and MUC3B was also expressed in basal cells. MUC3A and MUC3B mRNA levels were higher in current smokers, regardless of airway obstruction, than in never-smokers. In line with this, they correlated positively with exhaled carbon monoxide and negatively with expiratory air flows. Moreover, patients with emphysema had higher MUC3A protein expression in ciliated cells, compared to patients without emphysema.

Several studies conducted previously failed to detect MUC3 expression in normal adult and fetal respiratory tract by mRNA in situ hybridization and immunohistochemistry [35,36,37]. One unknown type of MUC3 was studied in the bronchial mucosa of COPD patients during exacerbation, but information on the antibodies and concentration was not included in the publication [38]. Here, we showed that MUC3A and MUC3B proteins, detected by immunohistochemistry, were expressed in the large airway epithelium of healthy subjects who had never smoked, as well as in healthy current smokers and COPD patients. The reason for the discrepant results between our study and others is likely due to the fact that our staining protocol and reagents, including antibodies, differed from those used in the previously published studies. We speculate that our method and using the current reagents may be a more sensitive way to detect MUC3 protein expression in human tissues than the previously used methods.

MUC3 was first described as a unique gene but was later separated into two, MUC3A and MUC3B [32,39]. MUC3B has a glycosylated subunit that is longer than that of MUC3A. In the present study, even though MUC3A and MUC3B mRNA levels were both associated with lower expiratory air flows, they behaved in slightly different ways, since they did not show the same pattern in the correlation and group comparison analyses. Moreover, MUC3A and MUC3B immunohistochemical expression in ciliated cells were not correlated. This indicates that MUC3A and MUC3B in large airway epithelium may have two different regulatory pathways, and thus may play different roles in human lung. Our results support the notion that MUC3A and MUC3B indeed are two different mucins, as was appreciated in previous studies [32,39].

In the present study, MUC3B mRNA levels measured by microarray in samples acquired by bronchial brushing were higher in smokers and showed correlations with total cell concentration in BALF. In line with this, MUC3B mRNA levels correlated negatively with the airway obstruction measure FEV_1_. Taken together, this indicated that MUC3B mRNA in this cohort may be a biomarker associated with variables indicative of worse prognosis. Similarly, the MUC3A mRNA level in samples acquired by bronchial brushing was negatively correlated with postbronchodilator FVC assessed by dynamic spirometry. In addition, patients with emphysema had higher MUC3A expression in ciliated cells, compared with patients without emphysema. Simultaneously, both MUC3A mRNA and MUC3B mRNA were positively correlated with exhaled carbon monoxide being higher in smokers than in never-smokers. It remains unclear whether MUC3A and MUC3B mRNA acted here solely as surrogate markers of smoking exposure, which in its turn contributes to the unfavorable inflammatory profile and worse lung function, or whether MUC3A and MUC3B mRNA may play causal roles. Future studies are warranted to elucidate this.

MUC3A and MUC3B mRNA levels were associated with exhaled carbon monoxide, inflammatory cell pattern in BALF, and lung function. The immunohistochemical MUC3A and MUC3B expressions did not show the same associations, which result is linked to the strong expression profiles of both proteins, since expressions of both MUC3A and MUC3B were quite extensive, and the intensity was scored as moderate or strong in most cases (Figure 1). Nevertheless, exhaled carbon monoxide, inflammatory markers in BALF, and lung function were associated with MUC3A and MUC3B at the mRNA, but not at the protein, level. The results of different kinds of expressions of protein and mRNA levels have been observed also in other studies since a study on lysosomal sulfatases in the airways of COPD patients revealed increased expression of mRNA and decreased expression of proteins [40]. 

A limitation of this study was the relatively small number of subjects and that we were lacking mRNA data for COPD ex-smokers, as it would have been interesting to see whether MUC3A and MUC3B mRNA levels differed between COPD patients who were ex-smokers and those who were current smokers. In addition, the study design was observational cross-sectional, which enabled us to only create hypotheses based on associations. Future studies with prospective design are needed to evaluate the potential value of mucins as biomarkers of long-term prognosis. 

The results of a recent study suggested that female patients with COPD may have a different disease phenotype than male COPD patients [41]. In the present study, female subjects had lower MUC3A and MUC3B mRNA levels in bronchial brush samples than male subjects. However, this difference could have been due to differences in smoking status between the male and female subjects who provided mRNA. Interestingly, all of the four biopsy samples where we are able to detect MUC3A protein expression in basal cells were retrieved from female study participants, and three of those were current smokers. Since the number of subjects in the present study was relatively small and the sex differences might be confounded by smoking, larger studies are needed to evaluate whether female subjects have sputum of different properties than male subjects.

Mucins have recently received significant attention as potential biomarkers. Primarily, mucins have become increasingly recognized as valuable biomarkers in a wide spectrum of cancers, including lung cancer [42,43,44,45,46,47,48,49,50,51,52]. Therefore, mucins present potential molecular targets for cancer therapy [53,54,55,56]. In addition, mucins have been implicated in inflammatory processes. For example, a recent study has found that MUC8 is highly secreted in saliva. This study also found support for the role of MUC8 in the context of inflammatory events and salivary stone formation [57]. In hypersensitivity pneumonitis patients, the T allele of the MUC5B gene predicts lower baseline FVC and its subsequent decrease [58]. Recently, six novel genome-wide significant signals for chronic sputum production have been identified, and these include a chromosome 11 mucin locus containing MUC2, MUC5AC, and MUC5B [59]. In line with these findings, sputum MUC5AC was increased in patients with steroid-untreated mild asthma, and it correlated with markers of airway type 2/eosinophilic inflammation [60]. It is currently being debated whether mucus hypersecretion might aggravate the pathophysiological vicious circle in airway diseases such as COPD and COVID-19, and the literature recommends that future research should focus on targeting mucus hypersecretion as a potential novel therapeutic approach [61,62]. In COPD, total mucin concentration in induced sputum was associated with smoking history, chronic bronchitis, disease severity, and risk of acute exacerbations [63]. A subsequent study showed that increased MUC5AC concentration in the airways was associated with COPD onset and progression [64]. In the studies [60,64], MUC5AC concentration was measured in induced sputum samples, whereas in the present study we assessed the immunohistochemical expression of MUC3A and MUC3B in biopsies of the large airway epithelium. Therefore, the studies are not directly comparable, but interestingly, never-smokers, current smokers with normal lung function, and COPD patients did not differ in MUC3A or MUC3B expression in the collected tissue biopsies. In a state of mucus hypersecretion associated with smoking and COPD, and in light of the results of the studies [60,64], differences in MUC3A and MUC3B expression between the study groups would be expected; however, this was not the case. MUC5AC is a secretory mucin with a relatively well-described role in the respiratory tract, while MUC3A and MUC3B are membrane-tethered mucins that have not yet been studied in the context of pulmonary diseases. This underlines the novelty of the present study and highlights its significance. In the present study, the MUC3A and MUC3B mRNA levels were higher in current smokers compared to never-smokers, and they correlated positively with exhaled carbon monoxide and negatively with expiratory air flows. Additionally, subjects with emphysema had higher MUC3A protein expression in ciliated cells, compared to subjects without emphysema. This indicates that MUC3A and MUC3B may be biomarkers of pulmonary disease and that more research is needed to elucidate the role of these mucins in the respiratory epithelium. 

## 4. Materials and Methods 

### 4.1. Study Subjects

The study design was observational cross-sectional. The subjects included in this study were part of the Karolinska COSMIC (Clinical & Systems Medicine Investigations of Smoking-related Chronic Obstructive Pulmonary Disease) cohort (www.ClinicalTrials.gov/ct2/show/study/NCT02627872 (accessed on 21 August 2023)), as described previously [65,66,67,68,69]. From the general Stockholm population, aged 45–65 years, through advertisements, we recruited healthy never-smokers (n = 40), current smokers with normal lung function (n = 40), and current smokers or ex-smokers with COPD with FEV_1_/FVC < 0.7 and FEV_1_ 50–100% predicted (n = 38). Exclusion criteria included atopy (defined as positive specific IgE test), asthma, treatment with oral or inhaled glucocorticoids or with antibiotics for a COPD exacerbation within the 3 months prior to study entry, and significant ischemic heart disease or arrhythmia. All the study subjects were recruited during the years 2007–2011.

Current smoking exposure was assessed by exhaled carbon monoxide measurement [70]. All subjects underwent clinical examination, spirometry, computed tomography (CT) scan, and bronchoscopy. Emphysema was evaluated on inspiratory CT scans, by two experienced reviewers (SN and ReKa), and was considered to be present when more than 5% of the whole lung parenchyma was occupied by emphysematous changes. Chronic bronchitis was patient-reported and defined as mucus production for a period of at least three months during at least two years in a row. The study was approved by the Stockholm Regional Ethical Board (Case no 2006-959-31/1; 27 October 2006), and all participants gave informed, written consent.

### 4.2. Bronchoscopy and Inflammatory Markers

Bronchoscopy was performed as previously described [71,72]. Bronchial mucosal biopsy specimens were taken by use of pulmonary biopsy forceps with smooth-edge jaws (Radial Edge^®^ Biopsy Forceps, Boston Scientific, Boston, MA, USA). Four to six biopsies were taken from each subject, and they were collected from lobar or segmental carinae of the upper lobes or the apical segment of the lower lobes. All biopsies were immediately formalin-fixed and embedded in paraffin. The tissue samples were stained with hematoxylin-eosin and the representativeness of all biopsies was evaluated. Two representative tissue blocks from each subject were selected for immunohistochemical analyses of MUC3A and MUC3B. EGFR immunohistochemical staining results published previously were used also in the present study [20]. The methods of biopsy preparation were performed as previously published [20].

Bronchial brush samples were taken from lobar or segmental bronchi in the lingula segment or the basal lower lobe segment of the left lung. Bronchial brushing was performed by cytology brush (Olympus, BC-202D-2010, Tokyo, Japan) by brushing approximately 10 times back and forth around the circumference. The procedure was repeated five times in various segments, using the same brush, but the brush was gently shaken off between the brushings. The brush samples were taken carefully and a traumatically, for the purpose of only harvesting epithelial cells and not causing bleeding. The brush samples were washed in PBS, and either aliquoted as pellets and frozen at −150 °C until their use in molecular analyses, or prepared as cytospin glasses and frozen at −80 °C until their use in immunocytochemistry analysis. 

Bronchoalveolar lavage was performed (5 × 50 mL in the middle lobe) and cells were collected as previously described [71,73]. Total cell concentration, macrophage, lymphocyte, and neutrophil percentage and CD4/CD8 T cell ratio were determined in BALF, as previously described [65,72,74].

### 4.3. Immunohistochemical Staining and Analysis of Immunohistochemical Protein Expression 

Four μm thick sections were cut, deparaffinized with xylene, and rehydrated in a descending ethanol series. The primary antibodies used in the immunostaining were designed for formalin-fixed paraffin-embedded tissues. The antibodies used are summarized in Table 3. MUC3A and EGFR were stained with DAKO REAL EnVision-kit (Dako, Glostrup, Denmark) and MUC3B with FLEX-kit from Dako. All stainings were made according to the manufacturer’s instructions. Before application of the primary antibodies, the sections were heated in a microwave oven in 10 mM citrate buffer, pH 6.0, for 10 min. After overnight incubation at +4 °C with the primary antibody, samples were incubated for 30 min at room temperature with a biotinylated secondary HRP rabbit/mouse antibody (Dako, Envision, or Flex). In all the immunostainings, the color was developed with diaminobenzidine, after which the sections were lightly counterstained with hematoxylin. Representative stainings are shown in Figure 1.

In the evaluation of immunohistochemical samples, cytosolic expression was considered significant. The intensity of immunostaining was assessed as zero (negative) to four (strong positive) and the extent of the positive staining was estimated from 0% to 100% in each cell type. The score for each antibody correlated total intensity with extent, resulting in a total score with a range between 0 and 400 [20,75,76]. The evaluation was performed blinded to the clinical information of the study subjects and by an experienced researcher (HM), as described previously [20].

### 4.4. MUC3A and MUC3B mRNA Quantification by Microarray

MUC3A and MUC3B mRNA levels in samples acquired by bronchial brushing were measured by microarray for a subset of individuals, as previously described [77]. RNA was isolated using the NucleoSpin^®^ miRNA kit according to the manufacturer’s instructions (Macherey-Nagel, Düren, Germany). RNA quality and quantity were assessed for concentration and purity by determining UV 260/280 and 230/260 absorbance ratios obtained by the Nanodrop ND-1000 spectrophotometer (Nanodrop, Wilmington, DE, USA). RNA integrity and size distribution were examined by gel electrophoresis on RNA Pico LabChips (Agilent Technologies, Palo Alto, CA, USA) processed on the Agilent 2100 Bioanalyzer. RNA was isolated and amplified using the Low Input Quick Amplification Kit (Agilent Technologies), according to the manufacturer’s protocol, and subsequent Cy3-CTP labeling was performed by using one-color labeling kits (Agilent Technologies). Clean-up of the labeled and amplified probes was performed (Zymo Research Corporation, Irvine, CA, USA). The size distribution and quantity of the amplified product was assessed by Nanodrop. Equal amounts of Cy3-labeled target were hybridized to Agilent human whole-genome 4x44K Ink-jet arrays containing a total of 41,000 probes corresponding to 19,596 entrez genes. Hybridizations were performed at 65 °C for 17 h at a rotation of 10 rpm. Arrays were scanned by using the Agilent microarray G2565BA scanner (Agilent Technologies) with Scan region: Agilent HD (61 × 21.6) and a resolution of 5 µm, TIFF: 16 bit, XDR: 0.10. Raw signal intensities were extracted with Feature Extraction v10.1 software (Agilent Technologies). Flagged outliers were not included in any subsequent analyses. Microarray datasets were normalized using the quantile normalization method according to Bolstad et al. [78]. No background subtraction was performed, and the median feature pixel intensity was used as the raw signal before normalization. All procedures were carried out using functions in the R package limma in Bioconductor [79,80].

Due to the lack of biospecimens (due to a freezer breakdown accident), the total number of mRNA samples was relatively small (Table 1), and we were completely lacking mRNA data on COPD patients who were ex-smokers. Study subjects who provided mRNA samples had lower FEV1 and higher residual volume, as compared with subjects who did not provide mRNA samples (Table 2).

### 4.5. Statistical Methods

All values are expressed as the mean ± SD. All statistical analyses were done using Tibco Statistica (Version 13) and Graph Pad Prism 9 software (GraphPad, La Jolla, CA, USA). The Kruskal–Wallis-test was used to compare three or more independent groups. For comparison of two independent groups, the unpaired two-tailed Mann–Whitney U Test was used. Correlations between two variables were assessed using the non-parametric Spearman’s rank correlation test. A value of *p* < 0.05 was considered statistically significant (* *p* < 0.05, ** *p* < 0.01, *** *p* < 0.001). 

## 5. Conclusions

In summary, the present study indicated that immunohistochemical MUC3A and MUC3B expressions were localized to ciliated cells in human large airway epithelium. The results also suggested that MUC3A mRNA and MUC3B mRNA levels in samples acquired from large airways by bronchial brushing were higher in current smokers than in never-smokers. Furthermore, MUC3A and MUC3B mRNA levels correlated positively with exhaled carbon monoxide and negatively with expiratory air flows, which indicates that these mucins may be associated with smoking. Thus, MUC3A and MUC3B may present either a surrogate marker of lung function or play a causal role in the pathophysiology of lung diseases. Future research should focus on exploring the potential roles of MUC3A and MUC3B in human airways in different states of health and disease.

## Figures and Tables

**Figure 1 ijms-24-13546-f001:**
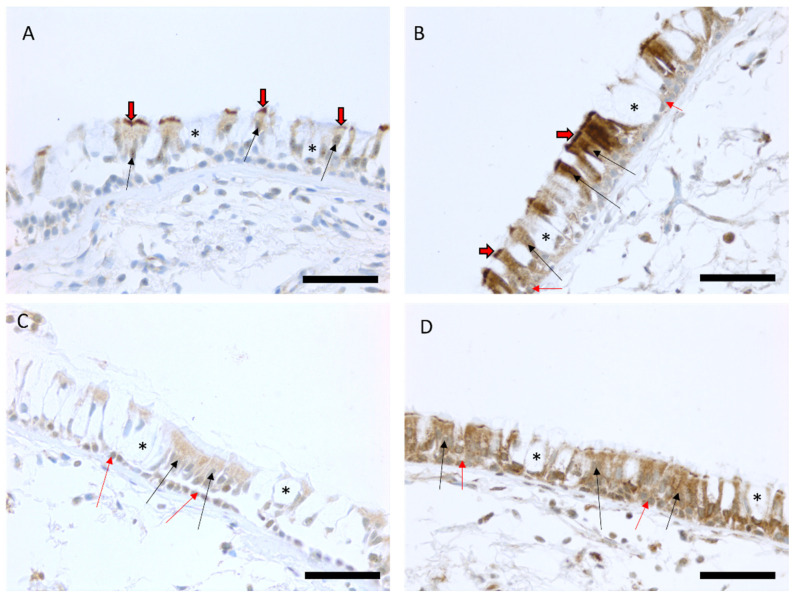
Representative images of the immunohistochemical stainings for MUC3A and MUC3B proteins in bronchial biopsy samples. Scores of the expression were assessed as negative, faint, moderate, strong, or very strong. Faint expression of MUC3A in the large airways of a smoker with COPD (**A**) and a strong MUC3A expression in the epithelium of a smoker with normal lung function (**B**). Faint MUC3B expression in ciliated and basal cells in a never-smoker (**C**) and a strong expression of MUC3B in epithelial cells in a never-smoker (**D**). Black arrows show the positive ciliated cells in the epithelium, black/red arrows point to the positive cilia of the cells, red arrows show the positive basal cells, and an asterisk is showing negative goblet cells. Scale bar is 50 µm.

**Figure 2 ijms-24-13546-f002:**
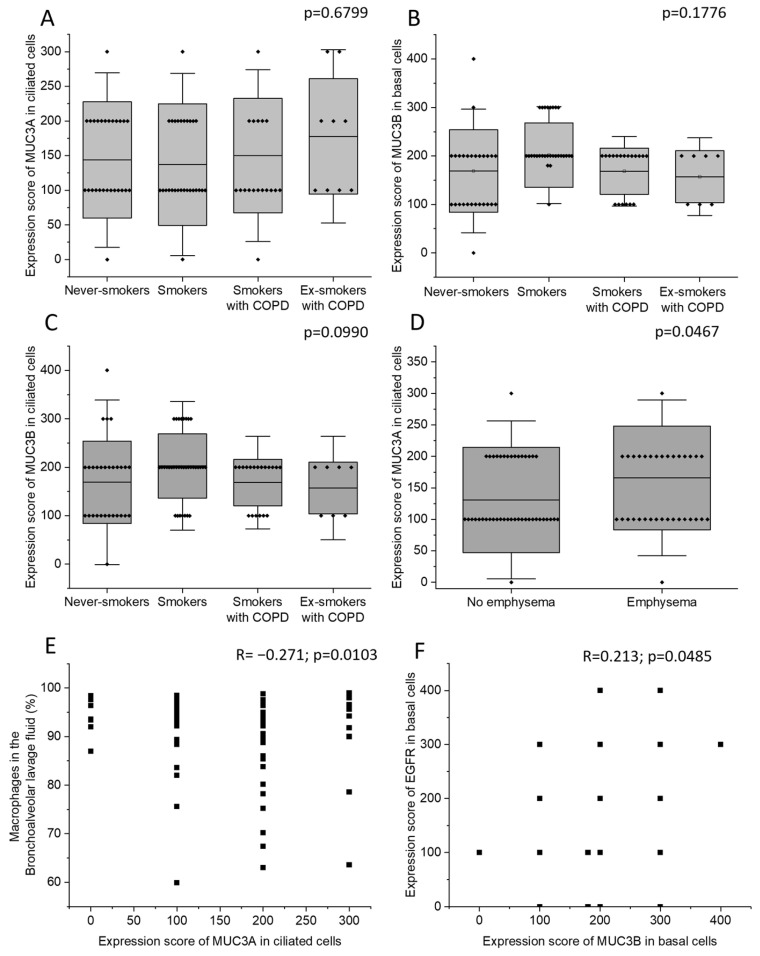
The immunohistochemical expression of MUC3A in ciliated cells (**A**), MUC3B in basal cells (**B**), and MUC3B in ciliated cells (**C**) in the four study groups: never-smokers, current smokers with normal lung function, currently smoking COPD patients, and COPD patients who were ex-smokers. The immunohistochemical expression of MUC3A in ciliated cells in subjects with and without emphysema (**D**). Correlation between the immunohistochemical expression of MUC3A in ciliated cells and macrophage percentage in BALF (**E**). Correlation between the immunohistochemical expression of MUC3B in basal cells and the immunohistochemical EGFR expression in basal cells (**F**). Bars represent mean and standard deviation. Kruskal–Wallis test in (**A**–**C**). Mann–Whitney U Test in (**D**). Spearman correlation analysis in (**E**,**F**). R: Spearman’s rank correlation coefficient.

**Figure 3 ijms-24-13546-f003:**
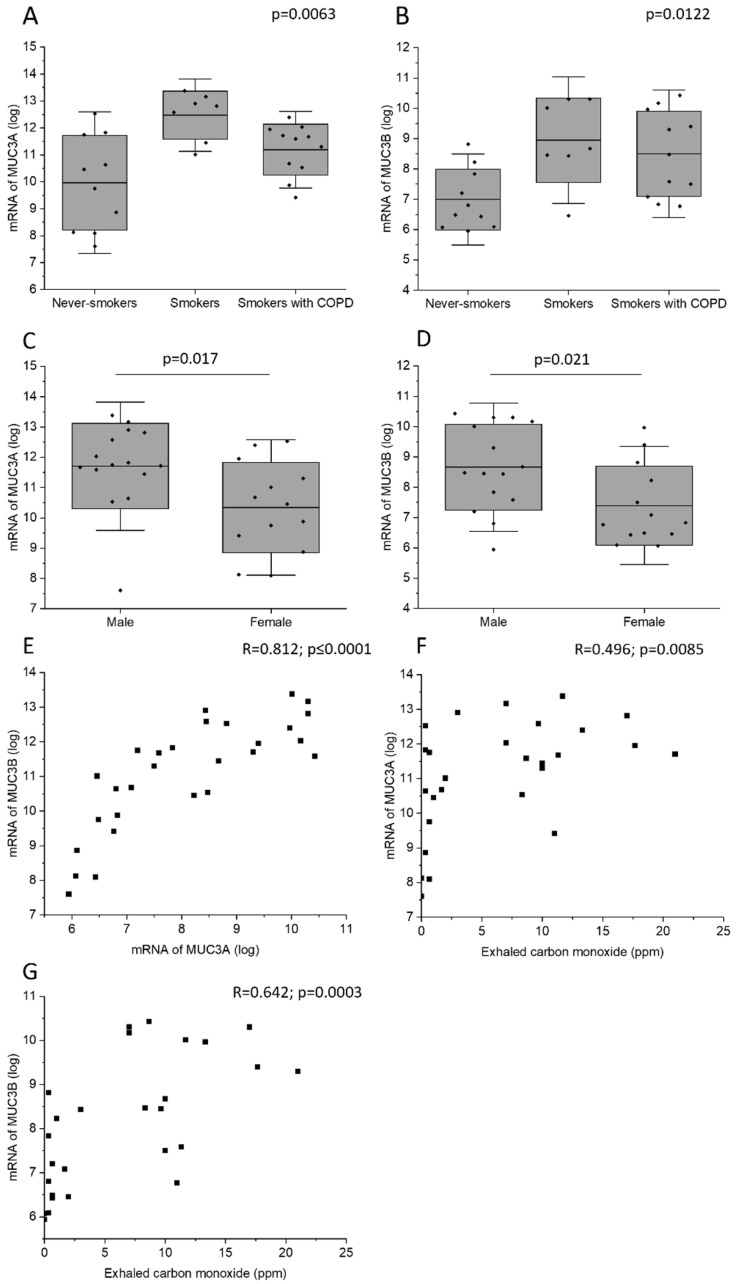
MUC3A mRNA (**A**) and MUC3B mRNA (**B**) in samples acquired by bronchial brushing across the study groups. Due to lack of biospecimens, mRNA samples from study subjects belonging to the study group of ex-smokers with COPD were missing. MUC3A mRNA (**C**) and MUC3B mRNA (**D**) in samples acquired by bronchial brushing in male and female study subjects. Correlation between MUC3A mRNA and MUC3B mRNA (**E**). Correlation between exhaled carbon monoxide and MUC3A mRNA (**F**) and MUC3B mRNA (**G**). Kruskal–Wallis test in (**A**,**B**). Bars represent mean and standard deviation. Mann–Whitney U Test in (**C**,**D**). Spearman correlation analysis in (**E**–**G**). R: Spearman’s rank correlation coefficient.

**Figure 4 ijms-24-13546-f004:**
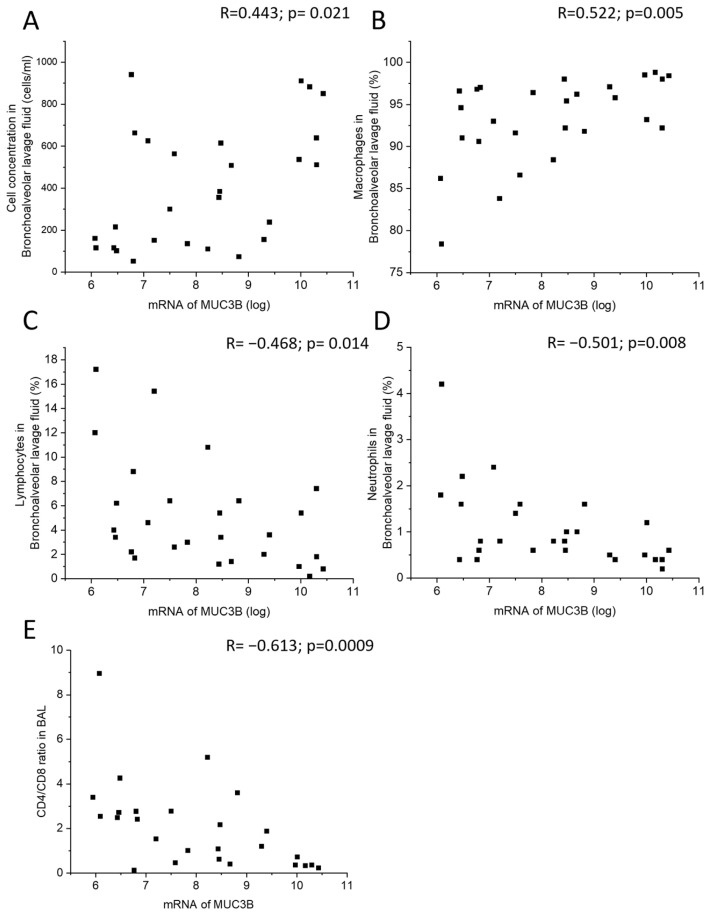
Correlation between MUC3B mRNA in samples acquired by bronchial brushing and total cell concentration in BALF (**A**), macrophage percentage (**B**), lymphocyte percentage (**C**), neutrophil percentage (**D**), and CD4/CD8 ratio in BALF (**E**). Spearman correlation analysis in all. R: Spearman’s rank correlation coefficient.

**Figure 5 ijms-24-13546-f005:**
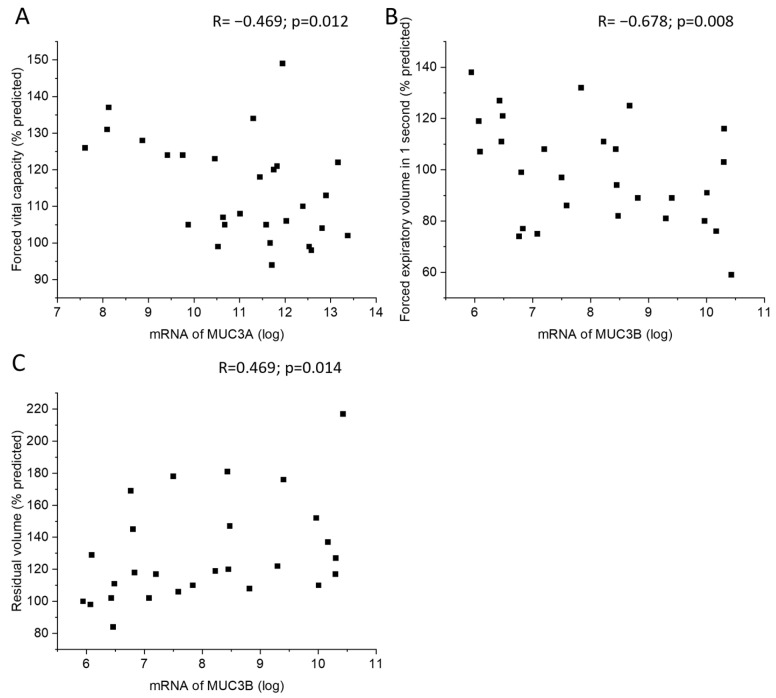
Correlation between MUC3A mRNA in samples acquired by bronchial brushing and postbronchodilator forced vital capacity in percent predicted (**A**). Correlation between MUC3B mRNA in samples acquired by bronchial brushing and postbronchodilator forced expiratory volume in 1 s in percent predicted (**B**) and residual volume in percent of predicted (**C**), counted according to ECCS reference equations. Spearman correlation analysis in all. R: Spearman’s rank correlation coefficient.

**Table 1 ijms-24-13546-t001:** Demographics of the study groups.

	Healthy Never-Smokers (n = 40)	Smokers (n = 40)	COPD Smokers (n = 27)	COPD Ex-Smokers (n = 11)
Age (years)	59.5 (51.0–63.8)	53.0 (49.0–58.6)	60.4 (56.0–63.0)	63.0 (54.1–65.0)
Female subjects (n (%))	20 (50)	20 (50)	12 (44)	6 (55)
Number of mRNA samples (n (%))	10 (25)	7 (18)	11 (41)	0 (0)
Current cigarette consumption (per day)	0 (0–0)	17 (12–20)	20 (11–20)	0 (0–0)
Smoking history (pack-years)	0 (0–0)	33.5 (28.5–40.0)	42 (36–48)	30 (20–38)
Exhaled carbon monoxide (ppm)	0.33 (0.00–0.84)	9.84 (6.67–11.50)	9.67 (2.67–13.33)	0.33 (0.00–5.33)
Postbronchodilator FEV_1_ (% predicted)	119.5 (109.5–127.0)	108.5 (103.0–116.5)	78.0 (74.0–85.0)	81.0 (65.0–88.0)
Postbronchodilator FVC (% predicted)	121.0 (110.5–128.0)	113.0 (106.5–124.0)	105.0 (94.0–110.0)	104.0 (95.0–116.0)
Postbronchodilator VC (% predicted)	120.5 (110.5–129.5)	114.0 (106.5–126.5)	106.0 (94.0–119.0)	111.0 (101.0–119.0)
Diffusing capacity (% predicted)	89.5 (84.0–96.5)	77.5 (72.0–85.0)	63.0 (60.0–74.0)	71.0 (49.0–77.0)
Residual volume (% predicted)	100.5 (91.5–111.0)	115.0 (94.0–127.0)	132.5 (107.0–165.0)	143.0 (124.0–160.0)
Emphysema (n (%))	1 (3)	22 (55)	21 (78)	8 (73)
Chronic bronchitis (n (%))	0 (0)	10 (25)	7 (26)	2 (18)

Data are presented as n or median (interquartile range). Percent predicted values were counted according to the ECCS reference equations. Abbreviations: FEV_1_: forced expiratory volume in 1 s; FVC: forced vital capacity; VC: slow vital capacity; COPD: chronic obstructive pulmonary disease.

**Table 2 ijms-24-13546-t002:** Comparison of the subgroups according to mRNA samples.

	Provided mRNA (n = 28)	Did Not Provide mRNA (n = 90)	*p*-Value
Female subjects (n (%), current smokers (n, %))	13 (46), 7 (54)	45 (50), 25 (56)	0.830, 0.192
Male subjects (n (%), current smokers (n, %))	15 (54), 11 (73)	45 (50), 24 (53)	0.830, 0.192
Age (years)	58.0 (50.0–62.9)	57.5 (52.0–63.0)	0.770
Current cigarette consumption (per day)	15 (0–20)	10 (0–17)	0.088
Smoking history (pack-years)	34 (0–41)	26 (0–40)	0.318
Exhaled carbon monoxide (ppm)	7.0 (0.7–11.0)	2.7 (0.3–10.0)	0.475
Postbronchodilator FEV_1_ (% predicted)	92.5 (80.0–108.0)	111.5 (92.0–121.0)	0.017
Postbronchodilator FVC (% predicted)	111.5 (104.5–124.0)	111.5 (103.0–123.0)	0.823
Postbronchodilator VC (% predicted)	111.5 (98.5–124.0)	114.0 (106.0–128.0)	0.168
Diffusing capacity (% predicted)	80.0 (63.0–92.0)	79.0 (70.5–88.0)	0.761
Residual volume (% predicted)	119 (108–147)	109 (94–129)	0.018
Emphysema (n (%))	12 (43)	40 (44)	1.000
Chronic bronchitis (n (%))	6 (21)	13 (14)	0.387

Data are presented as n or median (interquartile range). Percent predicted values were counted according to the ECCS reference equations. Mann–Whitney U Test for continuous variables. Fisher Exact Test for comparing proportions. Abbreviations: FEV_1_: forced expiratory volume in 1 s; FVC: forced vital capacity; VC: slow vital capacity.

**Table 3 ijms-24-13546-t003:** Antibodies used for the immunohistochemical staining.

	Producer	Kit	Antigen Retrieval	Dilution
MUC3A	Atlas Antibodies	Envision	Citrate pH 6	1/150
MUC3B	Abgent (C-term E881)	Flex	Citrate pH 6	1/1000
EGFR1	Novocastra. NCL-L-EGFR_384	Envision	Citrate pH 6	1/100

## Data Availability

The data presented in this study are available on request from the corresponding author. The data are not publicly available due to privacy or ethical restrictions.

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
