# Peer review of "Mucins 3A and 3B Are Expressed in the Epithelium of Human Large Airway"

_ijms, 2023, doi:10.3390/ijms241713546_

Round 1

Reviewer 1 Report

In the original research manuscript entitled “Mucins 3A and 3B are expressed in the epithelium of human large airway”, the authors measured MUC3A and MUC3B expression in COPD patients, smokers, and non-smoker subjects. Using immunostaining and microarray, they observed that both genes are expressed in ciliated cells. MUC3B is also detectable in basal cells. Patients with emphysema had higher MUC3A levels. Both MUC3A and MUC3B were higher in smokers compared to the non-smoker group suggesting these gene expressions in large airways may reflect the severity of the disease and play a role in the development of airway obstruction. In general, this is an excellent study filling a knowledge gap of MUC3A and MUC3B expression in COPD. The experiments are well designed, and the conclusions are strongly supported by results. I believe this work could substantially contribute to the research in the field.

Author Response

Please see the uploaded Word-file.

Reviewer 2 Report

Merikalio and colleagues present a highly relevant paper on the impact of mucins in lung diseases. The content is qualitative and the research questions are valuable.

There are some issues which should be included:

1- the quality of the figures are low

2- please explain in more detail how the IHC were performed and why this method is important. 
3- please extend the limitation section

4- please include some important manuscripts published between 2020-2022 on mucins regardings lung and other diseases and highlight the potential aspect of mucins as biomarkers

The language style is good.

Author Response

Please see the attached Word-file.

Reviewer 3 Report

·         In abstract, please state clearly the country/place where the study was conducted, the date of sampling took place, the type of study design (e.g., experimental, case-control, cross-sectional), the age of participants, the type of statistical analysis used and the significant p-value.

·         The introduction is short and very weak in its current format. Only 8 references were used. It is important that this section incorporates the following: (1) the prevalence and incidence of COPD in smokers and nonsmokers; (2) The causes of COPD other than smoking; (3) the molecular biology of COPD. How does the respiratory system differ in a person with COPD? Is COPD the same as emphysema, chronic bronchitis and asthma; (4) describe the mechanisms (if any) of how smoking affects COPD in smokers and nonsmokers; (5) Biomarkers should be better described and presented in the context of not only COPD but also other respiratory diseases such as asthma.   Mucin is not only the biomarker of COPD and asthma that influenced by smoking. Can COPD be caused by secondhand smoke? Are COPD and asthmatic smokers different from non-smokers in terms of biomarkers? I would suggest referring to these articles (ERJ Open Res. 2020 Jul 20;6(2):00192-2019; Curr Issues Mol Biol. 2023 Jun 10;45(6):5099-5117); and (6) the novelty of this research could be improved. Why this research is important in light of other studies? What is new? What this study adds? Why this research focused on mucin only? This should be clarified at the end of introduction.

·         This is confusing. Materials and methods should be placed as section 2 not 4.

·         Some details regarding the study design and measures are not reported. The study design is not clearly defined. How participants were recruited? What were the inclusion/exclusion criteria? Statistical analysis should be described in much more details.

·         Table 2 should be moved to results.

·         Discussion should be moved after results.

·         The discussion should be qualified with more references. It is important to support your results (e.g., see Lines 15-232) with recent references.

·         Is this study without limitations? I haven’t seen any limitation of the study.

·         Few references were very old- please delete/update (Ref # 4,8,10,11,12,15).

Moderate editing is required.

Author Response

Please see the attached Word-file.

Round 2

Reviewer 3 Report

Excellent improvement. No further comments.